# Translation and Cultural Adaptation of the StimQ for Use with Italian Children from Kindergartens

**DOI:** 10.3390/children10010109

**Published:** 2023-01-04

**Authors:** Roberta De Salve, Sara Romanelli, Francesco Frontani, Francesca Policastro, Anna Berardi, Donatella Valente, Giovanni Galeoto

**Affiliations:** 1Human Anatomy, Histology, Forensic Medicine and Orthopedics, Sapienza University of Rome, 00185 Rome, Italy; 2Departmental Faculty of Medicine and Surgery, Saint Camillus International University of Rome and Medical Sciences (UniCamillus), 00131 Rome, Italy; 3Department of Medical Science, University of Trieste, 34127 Trieste, Italy; 4Department of Human Neurosciences, Sapienza University of Rome, 00185 Rome, Italy; 5Neuromed, IRCCS, 86077 Pozzilli, Italy

**Keywords:** child development, validation, early brain development, environmental influences

## Abstract

The StimQ questionnaire is used to assess the home environment of children. The questionnaire is comprised of four subscales, and it was completed by the main caregiver. The items were different considering the band ages of the children: infants (5–12 months), toddlers (12–36 months), and preschoolers (36–72 months). The aim of the study was to translate, transculturally adapt, and evaluate the psychometric characteristics of the Italian version of the StimQ. To achieve this goal, a sample of 142 children was recruited from different kindergartens. The mean age of the group was 30.63 (SD 19.56), and 112 of them were female. The main caregiver was the mother in 95% of the cases. The Cronbach’s alfa was excellent, considering the total score (0.82 for infants, 0.85 for toddlers, and 0.86 for preschoolers). Intrarater reliability was performed by administering the questionnaire after 48 h and by two different researchers. Both analyses showed an excellent reliability for the total score and all the subscales. The intrarater reliability was 0.99 for the infant, 1 for the toddler and 0.99 for the preschooler age groups. The interrater reliability was 0.95 for the infant, 0.93 for the toddler, and 0.97 for the preschooler age groups. The StimQ is a reliable questionnaire that could be helpful for clinicians and researchers who work with children in Italy.

## 1. Introduction

The cognitive, motor, and linguistic development of a child is strictly related to social context and the activities proposed by caregivers [1,2,3]. The present literature highlights the importance of socioeconomic and cultural factors in this context, such as caregiver literacy, monthly income, and number of family members [4,5]. After birth, an infant’s early learning is the result of interaction with caregivers, who should immediately start talking, smiling, and singing to the baby [6,7]. Moreover, as suggested by the American Academy of Pediatrics, involving babies in reading and storytelling as soon as possible could support their cognitive and relational development [8,9]. For many years, the World Health Organization has encouraged reduced screen time, especially for younger children under 5 years of age. In fact, time spent watching or playing with technological devices reduces quality time with caregivers, is negatively associated with cognitive and physical development, and is positively associated with sleep problems, depression, anxiety, and obesity [10,11,12].

In Italy, the Cultural Association of Pediatricians developed many national programs to support quality interaction time between children and their caregivers, including “Nati per leggere” (“Born for Reading”) and “Nati per la musica” (“Born for Music”). However, currently, 43.5% of Italian children have a TV in their bedroom, and 44.5% spend more than two hours daily watching TV or playing with technological devices. Moreover, Italy has one of the highest obesity and sedentary rates in children age 6 to 9, compared to other European countries. These statistics highlight a need to define the parental model in order to support the cognitive and physical development of children and prevent developmental delays and disorders [13,14].

Considering the overuse of technological devices in the pediatric population and the common use of these devices in the parental model, especially to calm or distract children, introducing a parent-reported measure of the domestic cognitive environment in Italy is necessary. Dreyer et al. (1996) developed StimQ to support child development. It is a fact that the healthy development of a child is strictly related to the home environment, which emerges from the interactions of a wide range of factors, such as education, parent’s mental health, and economic circumstance [15,16]. StimQ is an ecological, reliable, and easy-to-use tool for assessing children under 6 years of age [17]. In the original article, the authors showed a good reliability of the total score (alpha = 0.88), reading (alpha = 0.9), availability of learning materials (ALM) (alpha = 0.71), and parental involvement in developmental advancement (PIDA) (alpha = 0.68), but a moderate reliability in regards to parental verbal responsivity (PVR) (alpha = 0.43). However, this tool is frequently used in the literature to investigate the home environment of children, especially in regards evaluating their access to stories (reading) [18], assessing parental responsiveness in pediatric primary care [19], verifying parent-child interactions [20], and identifying the home environment of hospitalized children [21]. Camp et al. [22] also reported that StimQ was a useful method to assess environmental factors that could influence early vocabulary development.

Due to the multiple potential applications of StimQ and its usability in clinical and research settings, the aim of the present study was to perform the translation, transcultural adaptation, and evaluation of the psychometric characteristics of the Italian version of StimQ. The adaptation and cross-cultural validation of an instrument is of great importance for producing a relevant cross-cultural analysis among the population, since the delivery of quality care depends on the accurate assessment and deeper understanding of an individual’s cultural, linguistic, and ethnic background [23]. To achieve our goal, we employed a native English language researcher, and an expert panel was created to check and synthetize the final draft of the questionnaire. After this process, we then analyzed the reliability of the questionnaire in order to make it useable for Italian clinicians and researchers working with young population. We expected to obtain similar results to those in the original article regarding the different psychometric characteristics considered.

## 2. Methods

This study was conducted by a research group at Sapienza University of Rome and the Rehabilitation and Outcome Measures Assessment (ROMA) Association, who were involved in different studies concerning rehabilitation [24,25,26,27,28,29,30,31,32,33,34].

## 3. Procedure

At the beginning of this study, we received consent from the developers of the original instrument to translate and adapt the StimQ questionnaire. For the translation and cultural adaptation of this questionnaire, we used the Principles of Good Practice for the Translation and Cultural Adaptation Process of Patient-Reported Outcomes Measures from Wild et al. [35]. The use of a translated, adapted, and evaluated version of StimQ for the Italian population in clinical and research settings is fundamental, as suggested by Beaton et al. [36]. To achieve these goals, we followed the Consensus-Based Standards for the Selection of Health Status Measurement Instruments (COSMIN) checklist of Mokkink et al. [37]. All procedures were followed in accordance with the ethical standards of the responsible committee on human experimentation (institutional and national) and in accordance with the 1975 Declaration of Helsinki, as revised in 2008. Ethics committee approval was not required for this study, since it involved the secondary use of clinical data that did not include any identifier that would allow attribution of private information to an individual. Informed consent was obtained from all participants included in the study.

## 4. Translation and Cultural Adaptation

The first step was a translation of the questionnaire from English to Italian. The translated StimQ was back-translated, and the translated version was compared with the original version. To adapt the final version of StimQ, we decided to create an expert panel, as suggested by the guideline. The panel was composed of two occupational therapists, a physical therapist, and a physician, representative of all the health professionals working with pediatric patients. The expert panel drafted the final version of StimQ.

## 5. Participants

We recruited participants from Italian kindergartens. The kindergartens were selected in the geographic area of Taranto. We contacted different kindergartens, and then we randomly chose those for inclusion in study from those that expressed interest in participating. In September 2021, the questionnaire was longitudinally administered in different kindergartens; this allowed for the recording of different statistical analyses regarding the reliability of the questionnaire. Since StimQ is a questionnaire that is completed by the main caregiver of the child, we included participants who were the main caregiver of a healthy child between the ages of 5 months and 6 years. We excluded Italians who were living outside Italy and families in which one of the parents was not originally from Italy. We contacted the parents of the children, and those who volunteered to participate on the study were enrolled. All participants completed an informed consent form that summarized the goal of the study. Participants were registered anonymously with the initials of their children. Considering the sample ages and concerns about the COVID-19 pandemic situation and restrictions, the study size included at least 30 participants for all the different age bands considered by the questionnaire. In the literature, the recommendation for the sample size ranges from 2 to 20 participants per item [38]. Shoukri et al. stated that under specific circumstances, only two or three replications per subject can be safely recommend [39].

## 6. Outcome Measures

StimQ is a questionnaire developed by Dreyer et al. (1996). This tool was developed to measure factors in the home environment that significantly correlate with a child’s cognitive developmental outcomes, including the quality of language stimulation, variety of objects and experiences in the child’s life, and maternal encouragement of and involvement in the child’s developmental advancements [17]. The questionnaire has been translated, adapted, and validated in two languages (Spanish and English) [17,40]. StimQ considers three different age groups: infants (5–12 months), toddlers (12–36 months), and preschoolers (36–72 months). For each age group, the questionnaire contains four subscales—reading (READ), parental involvement in developmental advancement (PIDA), parental verbal responsivity (PVR), and availability of learning materials (ALM)—with three subdimensions—quantity, quality, and diversity of concepts. The items considered in the questionnaire were different depending on the ages of the children. The administration of the questionnaire takes 15–20 min. The questionnaire consists of 35 items for infants, 33 for toddlers, and 41 for preschoolers. It is possible to analyze the subscale scores independently or by summing the scores obtained in each subscale. The score is different between the items. For many items, the score is dichotomic, and 1 point is provided for the answer yes and 0 for the answer no. For other items, mainly in the READ section, the score could range from 0 to 4, considering the amount of stimulus (for example: “Name some children’s books that you have at home and read to your child”). This allowed the clinician or to the researcher to focus their attention on just one of the aspects of the StimQ, or to consider the scale in its entirety. In the original article, Dreyer et al. (1996) stated that the internal consistency of the instruments ranged from 0.43 to 0.9, and that the intraclass reliability was from 0.82 and 0.93. The concurrent validity was measured, examining the associations between cognition, language, and socio-emotional outcomes at 36 months, and the score showed a correlation ranging from 0.17 to 0.4 [17].

## 7. Reliability

Internal consistency was defined as the extent to which items in a subscale are homogeneous, thus measuring the same concept [41]. Internal consistency of the PMI was examined by Cronbach’s alpha to assess the homogeneity of the scale by measuring the interrelatedness of items. Two subscale scores were calculated separately at baseline assessment. A Cronbach’s alpha value between 0.70 and 0.95 was considered satisfactory [41]. This property was analyzed for all the subsections of the subscales, and for the total score of the questionnaire and the subscales.

## 8. Intrarater and Interrater Reliability

The analysis of the intrarater reliability was performed by administering the questionnaire again after 48 h to a randomized subgroup of subjects. The sample number was established at 30 participants. The researcher did not explain the purpose of the second administration of the questionnaire to the selected subjects in order to prevent interference with the responses. The same researcher administered the questionnaire to the participants. Test-retest reliability was calculated using the ICC with a 95% confidence interval (CI). ICC values greater than 0.75 were the minimum requirement to use the instrument in group measurements [41].

Interrater reliability was assessed by using two different researchers to administer the StimQ to the participants. The subjects included in this analysis were blinded to the aim of this procedure. In our study, two researchers (RDS and SR) administered the questionnaire to a randomized subgroup of participants. The participant received the questionnaire from two different researchers, but with the same information about the compilation of the StimQ. The researchers did not furnish to the subjects with the reasons why they had to complete the questionnaire again, but they were available to address any other concern about the compilation tool. An ICC value higher than 0.7 was considered positive [41].

## 9. Statistical Analysis

All statistical analyses were performed using IBM^®^ SPSS^®^ (Statistical Package of Social Sciences) version 27 (Chicago, IL, USA). The Cronbach’s alpha was analyzed to investigate the reliability of the questionnaire. Moreover, two different subgroups were randomly created to analyze the interrater and intrarater reliability, which were investigated with the Cronbach’s alpha.

## 10. Results

### 10.1. Translation and Cultural Adaptation

The first step was a translation of the questionnaire from English to Italian. This step was performed by three researchers who were familiar with both languages, and then a blinded native English language researcher synthetized the three versions. The translated version of StimQ was back-translated by the three Italian researchers, and these three back-translated versions were integrated into one version that was compared with the original version of the tool. The focus group minimized the conceptual differences between the original version and the translated version so that all items in the translated version were as similar as possible to those in the original version.

### 10.2. Participants

The sample size required to achieve our research objectives was at least 30 children for each StimQ age group. Data collection was performed in September 2021 and was conducted anonymously and voluntarily for each participant. Informed consent was obtained from each participant in the study. The participant characteristics are summarized in Table 1.

The mean age of the sample was 30.63 months, with a standard deviation of 19.56. There is no gender prevalence, as 112 on 212 children were female (52.8%). The main caregiver for all three band ages was the mother, with an overall score of 95%. The average number of family members for the sample was of 3.83, with an SD of 0.77.

### 10.3. Reliability

The internal consistency was analyzed using Cronbach’s alpha. The total score ranged from 0.82 to 0.86 on the StimQ and from 0.74 to 0.91 on for StimQ core + ALM. A low internal consistency was obtained for the PIDA and PVR subscales for infants and toddlers and only for PVR for the preschoolers. The results are shown in Table 2, Table 3 and Table 4.

### 10.4. Intrarater and Interrater Reliability

For all subgroups, a sample of 30 participants was blinded and anonymously analyzed to assess intrarater and interrater reliability. Interrater reliability showed an excellent ICC for all age groups. Scores ranged from 0.97 (CI: 0.87–0.99) to 1. Intrarater reliability was good to excellent. The PIDAfor toddlers achieved a good score (ICC 0.67, CI: 0.31–0.91), while all other subscales showed excellent intrarater reliability. The results are presented in Table 5, Table 6 and Table 7.

## 11. Discussion

Home environment activities are important in order to achieve the proper development of a child [18,42]; for this development, the parental and main caregiver interaction are fundamental [20,43]. The StimQ is a questionnaire which aims to evaluate these characteristics. The goal of our study was to translate, adapt, and assess the psychometric characteristics of this tool for an Italian population. First of all, we performed a translation and transcultural adaptation of the questionnaire. To achieve these goals, the StimQ was translated and back-translated by native English speakers. The draft version was analyzed by a draft panel in order to reduce and transculturally adapt the questionnaire. When the final draft was ready, different psychometric properties were investigated. The reliability of the questionnaire was excellent, with Cronbach’s alpha values of 0.82 for infants, 0.85 for toddlers, and 0.86 for preschoolers. The PVR subscale did not reach a good level of reliability in all the band ages considered in the questionnaire (the Cronbach’s alpha was 0.53 for infants, 0.45 for toddlers, and 0.64 for preschoolers). This subscale considers the responsivity of the main caregiver and his/her capability to interact with the child. Considering the voluntarily aspect of this study, we observed that participants were critical of their behavior and, after completing the questionnaire, frequently asked if their behavior was appropriate. This could potentially explain the low level of reliability of this subscale. However, as expected, we obtained similar results to those in the original article; in fact, these data were coherent with those in the literature, e.g., in the original article, the PVR reached a Cronbach’s alpha of 0.43. The authors suggested that the small number of items which composed the subscales of PVR could explain a lower score in the Cronbach’s analysis. The reliability of the PIDA was low for infant and toddlers (0.43 and 0.28), but not for preschooler (alpha = 0.84). Regarding this subscale, there are different numbers of items; in fact, for the infant and toddler age groups, there are only 5 items, while in the preschooler age group, there are 12 items. This element could cause the differences in the results. The READ and ALM subscales reached a Cronbach’s alpha ranging from 0.70 to 0.88, showing an excellent reliability. Despite these controversial results, the reliability of the results of the total score of StimQ + ALM was excellent. The Cronbach’s alpha of StimQ + ALM was excellent for all age groups (0.74 for infants, 0.91 for toddlers, 0.87 for preschoolers), demonstrating a significant reliability for this questionnaire, when the analysis considered the total score. Considering the StimQ core, the reliability was excellent, with 0.82 for the infants, 0.85 for the toddlers, and 0.86 for the preschoolers. The test-retest score was excellent for all age groups, subscales, and total scores of StimQ. These ranged from 0.94 (CI 0.76–0.98) to 1, demonstrating excellent reliability at 48 h. This result is coherent with those in the original article, in which the authors found an intraclass correlation coefficient of 0.93 (CI 0.6–0.92) [17]. Interrater reliability is an important psychometric characteristic of a questionnaire. This parameter allows for the evaluation of the usability of a research and clinical instrument [44]. StimQ demonstrated an excellent ICC score, which ranged from 0.76 (CI 0.05–0.94) to 1. The only exception was the PIDA score for toddlers, for which the ICC score was 0.67 (CI 0.31–0.91), which was slightly lower than 0.75. However, this result did not change the excellent total score regarding StimQ or StimQ core + ALM. Intrarater reliability is an important characteristic that makes the questionnaire useful for clinical and research aims. StimQ showed excellent results in terms of psychometric characteristics, indicating that StimQ could be helpful in Italian clinical practice and research to identify families who do not adequately stimulate their children. In the future, comparing the results considering the different socio-economic levels of the families, or the number of family members, could be interesting, because the socio-demographic aspects could influence the home environment and the development of the children.

## 12. Limitations

This was a pilot study to determine some of the psychometric characteristics of StimQ. Construct validity should be further analysed in future studies identifying an appropriate tool for comparison. For the infant section, the sample size was more than predicted but could be considered small; however, this is a pilot, and we would increase the numerosity in the near future. Investigating the socioeconomic status of the family could be interesting to assess differences between the children development. A validated tool for the construct validity was not retrieved, for this, this analysis was not performed.

## 13. Conclusions

The StimQ was assessed on an Italian sample, and the questionnaire achieved excellent reliability for the total score (Cronbach’s alpha = 0.82 for infants, 0.85 for toddlers, and 0.86 for preschooler) and for StimQ core + ALM (alpha = 0.74 for infants, 0.91 for toddlers, and 0.87 for preschoolers). Interrater and intrarater reliability achieved an excellent result for the total score of the questionnaire, considering that all the total scores reached a value higher than 0.9. Only the PIDA subscale in the toddlers reached a low interrater value (0.67). The other subscales obtained values higher than 0.75 for all the band ages considered by the questionnaire. The StimQ is a reliable questionnaire that could be helpful for clinicians and researchers who work with children in Italy. This study provides essential information for both research and clinical practice. Clinicians now have a method to measure this invisible barrier, and they will be able to make informed decisions when setting up a treatment plan. The STIMQ also provides researchers with a useful tool in an important and relevant area of study for future research.

## Figures and Tables

**Table 1 children-10-00109-t001:** Characteristics of the different age groups considered in the study.

	Infant n°57	Toddler n°86	Preschooler n°69
Age, months (mean ± SD)	8.12 ± 2.2	23.14 ± 7.2	52.3 ± 9.3
Female gender n° (%)	33 (57.9)	42 (48.8)	37 (53.6)
Caregiver mother n° (%)	56 (98)	80 (93)	66 (95.6)
Number of family members n° (%)	3.9 (0.79)	3.77 (0.92)	3.85 (0.65)

**Table 2 children-10-00109-t002:** Cronbach’s alpha of subscale and total scores for infants.

STIMQ	Subscale	Mean ± SD	Cronbach’s Alpha
READ	TOTAL	5.05 ± 4.52	0.87
PIDA	TOTAL	3.47 ± 1.23	0.43
PVR	TOTAL	13.09 ± 2.36	0.53
ALM	TOTAL	5.86 ± 2.39	0.86
StimQ Core		21.61 ± 6.1	0.82
StimQ Core + ALM		27.5 ± 7.4	0.74

**Table 3 children-10-00109-t003:** Cronbach’s alpha of subscale and total scores for toddlers.

STIMQ	Subscale	Mean ± SD	Cronbach’s Alpha
READ	TOTAL	9.7 ± 4.48	0.85
PIDA	TOTAL	3.66 ± 0.90	0.28
PVR	TOTAL	12.37 ± 1.85	0.45
ALM	TOTAL	10.02 ± 7.17	0.88
StimQ Core		25.73 ± 5.9	0.85
StimQ Core + ALM		35.76 ± 11	0.91

**Table 4 children-10-00109-t004:** Cronbach’s alpha of subscale and total scores for preschoolers.

STIMQ	Subscale	Mean ± SD	Cronbach’s Alpha
READ	TOTAL	12.5 ± 3.9	0.84
PIDA	TOTAL	10.66 ± 3.42	0.84
PVR	TOTAL	15.21 ± 2.57	0.64
ALM	TOTAL	7.69 ± 0.6	0.70
StimQ Core		38.14 ± 7.12	0.86
StimQ Core + ALM		45.86 ± 7.37	0.87

**Table 5 children-10-00109-t005:** Interrater and intrarater reliability scores for infants.

STIMQ	Intrarater			Interrater		
	Test Mean ± SD	Re-Test Mean ± SD	ICC [95% CI]	Researcher 1 Mean ± SD	Researcher 2 Mean ± SD	ICC [95% CI]
READ	4.56 ± 4.65	3.30 ± 4.45	0.99 [0.99–1]	4.56 ± 4.65	3.50 ± 4.77	0.76 [0.05–0.94]
PIDA	3.35 ± 1.18	2.60 ± 0.97	0.97 [0.89–0.99]	3.35 ± 1.18	2.50 ± 1.18	0.99 [0.96–0.99]
PVR	12.74 ± 2.12	11.60 ± 2.46	0.94 [0.76–0.98]	12.74 ± 2.12	11.80 ± 2.35	0.85 [0.42–0.96]
ALM	5.32 ± 0.81	5.20 ± 0.79	1	5.32 ± 0.80	5.30 ± 0.82	0.85 [0.43–0.96]
StimQ core	20.65 ± 5.49	17.50 ± 4.30	0.99 [0.96–0.99]	20.65 ± 5.49	17.80 ± 4.94	0.95 [0.80–0.98]
StimQ core + ALM	25.97 ± 5.7	22.70 ± 4.11	0.99 [0.96–0.99]	25.97 ± 5.69	23.10 ± 4.93	0.92 [0.71–0.98]

**Table 6 children-10-00109-t006:** Interrater and intrarater reliability scores for toddlers.

STIMQ	Intrarater			Interrater		
	Test Mean ± SD	Re-Test Mean ± SD	ICC [95% CI]	Researcher 1 Mean ± SD	Researcher 2 Mean ± SD	ICC [95% CI]
READ	12.4 ± 1.71	12.4 ± 1.71	1	12.4 ± 1.71	13 ± 1.49	0.95 [0.80–0.98]
PIDA	4.3 ± 0.95	4.3 ± 0.95	1	4.3 ± 0.95	4.3 ± 0.67	0.67 [0.31–0.91]
PVR	13.5 ± 1.18	13.5 ± 1.18	1	13.5 ± 1.18	13.8 ± 1.55	0.90 [0.60–0.97]
ALM	8 ± 1.25	8 ± 1.25	1	8 ± 1.25	7.9 ± 1.29	0.94 [0.78–0.98]
StimQ core	30.2 ± 3.08	30.2 ± 3.08	1	30.2 ± 3.08	31.1 ± 2.47	0.93 [0.74–0.98]
StimQ core + ALM	38.2 ± 3.99	38.2 ± 3.99	1	38.2 ± 3.99	39 ± 3.3	0.90 [0.92–0.99]

**Table 7 children-10-00109-t007:** Interrater and intrarater reliability scores for preschoolers.

STIMQ	Intrarater			Interrater		
	Test Mean ± SD	Re-Test Mean ± SD	ICC [95% CI]	Researcher 1 Mean ± SD	Researcher 2 Mean ± SD	ICC [95% CI]
READ	10.8 ± 6.1	10.8 ± 6.1	1	10.8 ± 6.1	10.6 ± 5.98	0.99 [0.96–0.99]
PIDA	8.9 ± 4.22	8.7 ± 4.06	0.99 [0.96–0.99]	8.9 ± 4.22	8.9 ± 4.4	0.94 [0.76–0.98]
PVR	14.4 ± 2.95	14.5 ± 2.8	0.97 [0.98–0.99]	14.4 ± 2.95	14.5 ± 2.9	0.92 [0.68–0.98]
ALM	7.9 ± 0.3	7.9 ± 0.3	1	7.9 ± 0.3	7.9 ± 0.3	1
StimQ core	34.1 ± 10.34	34 ± 9.8	0.99 [0.99–1]	34.1 ± 10.34	34 ± 10.19	0.97 [0.88–0.99]
StimQ core + ALM	42 ± 10.43	41.9 ± 9.9	0.99 [0.99–1]	42 ± 10.43	41.8 ± 10.23	0.97 [0.88–0.99]

## Data Availability

All the data generated or analyzed during this study are included in this published article.

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
