# Peer review of "Translation and Cultural Adaptation of the StimQ for Use with Italian Children from Kindergartens"

_children, 2023, doi:10.3390/children10010109_

Round 1

Reviewer 1 Report

Dear authors,

As a general observation: the article does not comply with the journal's presentation pattern (introduction, methods, results, discussion, conclusions, references). Considering the existing structure, it looks more like a research report. Therefore, it requires restructuring for an article presentation.

Considering the purpose of the article, it must be emphasized why this moment, of the translation and cultural adaptation of a questionnaire, is important in the process of applying a standardized instrument. This must be reflected both in of Discussion and Conclusions sections.

Author Response

We are thankful for you suggestion about our manuscript. We tried to improve the study as you suggested. You find in the manuscript and in the following line the answer to your review request.

As a general observation: the article does not comply with the journal's presentation pattern (introduction, methods, results, discussion, conclusions, references). Considering the existing structure, it looks more like a research report. Therefore, it requires restructuring for an article presentation.

About the structure of the manuscript we tried to follow the guidelines of the journal. In the template they stated to use the presentation pattern as introduction, methods, results, discussion and conclusions. We added a section related to procedure on the methods section.

Considering the purpose of the article, it must be emphasized why this moment, of the translation and cultural adaptation of a questionnaire, is important in the process of applying a standardized instrument. This must be reflected both in of Discussion and Conclusions sections.

As suggested, we deeply modified all the section of the study in order to highlight the paper on the adaptation and translation process and on the analysis of the psychometric properties of the questionnaire. In the revised manuscript you will find all the revision made in accordance with your suggestions.

Reviewer 2 Report

This is a review of the manuscript titled "Translation and cultural adaptation of StimQ2 in the Italian 2 population". The aim of this study is to develop an Italian version of the StimQ and evaluate its psychometric characteristics. My concerns for this manuscript are as follows:

1. One major concern of this study is its small sample size. It is stated that, "Participants from Italian kindergartens have been recruited and 30 accepted to participate." (p. 1) This sample size is insufficient for evaluating the psychometric properties of an instrument.

2. The authors stated that the Italian StimQ is a "valid and reliable questionnaire" (p. 1 and 7). However, this study did not examine the validity of the StimQ.

3. There is an incorrect line break on p. 1, line 32.

4. It is stated that "Dreyer et al. (1996) developed StimQ to support child development" (p. 2). Should the StimQ be an instrument measuring home environment?

5. The authors should provide more descriptions of the original StimQ in the Introduction, such as its different versions and its subscales. Moreover, the psychometric properties of the StimQ documented in previous studies should be introduced in more detail.

6. More details about the StimQ should be described in the Method section. For example, what was the response format for each item?

7. The authors state that the StimQ was "completed by the main caregiver of the child" (p. 2). It is not clear why interrater reliability was evaluated. How raters were involved?

8. It is not clear why test-retest reliability was evaluated. The authors did not mention that the questionnaire was administered longitudinally.

9. Many Cronbach’s alpha coefficients were below .50 (p. 4-5). These findings were not consistent with the authors’ conclusion that the Italian StimQ was reliable.

Author Response

We are thankful for your revision, and we tried to improve all our work following your indication. In the following part you will find the point-by-point reply.

  1. One major concern of this study is its small sample size. It is stated that, "Participants from Italian kindergartens have been recruited and 30 accepted to participate." (p. 1) This sample size is insufficient for evaluating the psychometric properties of an instrument.

We agreed about this consideration, we try to add information about the decision of the sample size. We know that the numerosity of the sample for the infant was small and we add this consideration in the limitation section.

  1. The authors stated that the Italian StimQ is a "valid and reliable questionnaire" (p. 1 and 7). However, this study did not examine the validity of the StimQ.

 As you correctly stated, we did not assess the validity so we removed valid in both points.

  1. There is an incorrect line break on p. 1, line 32.

 As suggested, it was edited.

  1. It is stated that "Dreyer et al. (1996) developed StimQ to support child development" (p. 2). Should the StimQ be an instrument measuring home environment?

As suggested, we added information about how much is important the home environment for the child development. We totally agree with you about the goal of the StimQ, we intended that these two elements are deeply related so assessing accurately one of this (home environment) we could support the child development.

  1. The authors should provide more descriptions of the original StimQ in the Introduction, such as its different versions and its subscales. Moreover, the psychometric properties of the StimQ documented in previous studies should be introduced in more detail.

 As suggested, we described on the introduction section the psychometric properties showed in the original article. Moreover, we added information about the questionnaire on the outcome measure section.

  1. More details about the StimQ should be described in the Method section. For example, what was the response format for each item?

 As suggested, we provided more information about the questionnaire on the outcome measure section.

  1. The authors state that the StimQ was "completed by the main caregiver of the child" (p. 2). It is not clear why interrater reliability was evaluated. How raters were involved?

 As suggested, we added information about the administration and fulfilling of the questionnaire for the analysis of the interrater reliability.

  1. It is not clear why test-retest reliability was evaluated. The authors did not mention that the questionnaire was administered longitudinally.

 As suggested, we added information about the administration of the questionnaire in the participant section.

  1. Many Cronbach’s alpha coefficients were below .50 (p. 4-5). These findings were not consistent with the authors’ conclusion that the Italian StimQ was reliable.

We agree with your consideration. We added information about this on discussion section. In fact, also for the original article (Dreyer et al. 1996) it was found a low score for the PVR subscale, moreover the author did not analyse each subsection of the subscale but only the total score for the subscale. As you can see from the revised discussion, we added a comparison between our results and the originals, and the possible reason of the low score of Alpha. Of course, if you think that is better to remove the analysis of the subsection of the questionnaire, we are going to modify the table.

Reviewer 3 Report

This paper has as aim to perform the translation, transcultural adaptation and evaluation of the psychometric characteristics of the Italian version of StimQ. I believe that the authors need to address a few major and some minor issues to strengthen their argument and improve readability of the article. Below I make some suggestions for changes that could further improve this report.

---Specific comments---

1.      I recommend included the age population on the tittle. As authors said in the method section “Italian kindergartens”. In the tittle, I believe the word “StimQ2” is misspelled. Moreover, there is an error in the subscripts of the authors and affiliations

2.      Abstract section should include a little description of the questionnaire, more specific sample information (e.g., Mean, SD, % women and % men…) and the statistical analyzes carried out. Authors could review these aspects in the section.

3.      Keywords is missing. Please, add 4-5 keywords after abstract section.

4.      The introduction section is underdevelopment. The authors correctly justify the need to validate the scale in the Italian population. However, a section in the introduction where authors explain in detail the measure on which the study is based should be included. The authors explain part of this information inside of method section. I recommend reorganized the manuscript and add this section at the end of the introduction.

5.      The present study section is missing in the manuscript. Please add this section, make sure all objectives and hypotheses are stated in this section and explain the results expected.

6.      Please describe the characteristics of the sample in more detail in terms of socioeconomic status.

7.      The procedure is explained without section name. The paragraph after “method” should be renamed as “procedure”. How was the study conducted and how was the sample accessed? After read this section these points are not clear to me. Please revise the entire section.

8.      Statistical analysis is not explained in the section. This section should explain in detail what data analysis have been carried out and in what order.

9.      Discussion section is quite brief and underdeveloped. I suggest that the authors review this section and make an effort to connect it with the information provided throughout the manuscript according to the changes that I suggest in previous sections.

10.   Please, explain the limitations in more detail. The small sample could be added in this section. Moreover, strengths of the manuscript should be explained too.

11.   At the end of the manuscript, the practical implications and contributions of the study should be explained in details.

12.   Please, revise according to 7th Edition APA style all manuscript. Citing and referencing programs commonly used in statistical analysis (e.g., SPSS) is no longer necessary.

Author Response

We are thankful for your revision, and we tried to improve all our work following your indication. In the following part you will find the point-by-point reply.

  1. I recommend included the age population on the tittle. As authors said in the method section “Italian kindergartens”. In the tittle, I believe the word “StimQ2” is misspelled. Moreover, there is an error in the subscripts of the authors and affiliations

As suggested we changed the title of the paper.

  1. Abstract section should include a little description of the questionnaire, more specific sample information (e.g., Mean, SD, % women and % men…) and the statistical analyzes carried out. Authors could review these aspects in the section.

As suggested, we modified the abstract following your instructions and considering the request from the journal.

  1. Keywords is missing. Please, add 4-5 keywords after abstract section.

As suggested we added 4-5 keywords

  1. The introduction section is underdevelopment. The authors correctly justify the need to validate the scale in the Italian population. However, a section in the introduction where authors explain in detail the measure on which the study is based should be included. The authors explain part of this information inside of method section. I recommend reorganized the manuscript and add this section at the end of the introduction.

As suggested, we modified the last part of the introduction including more information about the detail and measure that were used in the study. Moreover, we tried to develop the introduction overall adding general information about the StimQ and other references for the children development.

  1. The present study section is missing in the manuscript. Please add this section, make sure all objectives and hypotheses are stated in this section and explain the results expected.

As suggested, we modified the last part of the introduction in order to add information about expected results and aim of the study. We did not create a specific section because it was not suggested by the format of the journal. If you prefer, we are going to create a specific section after the introduction including all this information.

  1. Please describe the characteristics of the sample in more detail in terms of socioeconomic status.

As suggested, we added in the participant section of the results more information about the sample group. Unfortunately, we did not investigate the socioeconomic status of the family but just the numerosity. So we add this point as a limitation.

  1. The procedure is explained without section name. The paragraph after “method” should be renamed as “procedure”. How was the study conducted and how was the sample accessed? After read this section these points are not clear to me. Please revise the entire section.

As suggested, we modified the participants section including the information about the way the sample was accessed and how they were investigated.

  1. Statistical analysis is not explained in the section. This section should explain in detail what data analysis have been carried out and in what order.

As suggested, we modified the section statistical analysis providing more information about the data analysis

  1. Discussion section is quite brief and underdeveloped. I suggest that the authors review this section and make an effort to connect it with the information provided throughout the manuscript according to the changes that I suggest in previous sections.

As suggested, we tried to improve this section; you can find all the revisions in the discussion section.

  1. Please, explain the limitations in more detail. The small sample could be added in this section. Moreover, strengths of the manuscript should be explained too.

As suggested, we modified this section considering the sample and other limitations.

  1. At the end of the manuscript, the practical implications and contributions of the study should be explained in details.

We added this in the discussion section.

  1. Please, revise according to 7th Edition APA style all manuscript. Citing and referencing programs commonly used in statistical analysis (e.g., SPSS) is no longer necessary.

As suggested, we changed the references according to your request.

Round 2

Reviewer 1 Report

Dear authors,

Please review the first part of the abstract. The word introduction should be deleted, and the first sentence should be found after the article's purpose.

Also, please develop the Conclusions part more.

Reviewer 2 Report

This is a review of the revised manuscript titled "Translation and cultural adaptation of StimQ2 in the Italian 2 population". This manuscript has been improved. However, the issue of very small sample size cannot be fixed without collecting additional data. The current sample size is insufficient for evaluating the psychometric properties of an instrument.

Round 3

Reviewer 2 Report

The manuscript has been improved and is acceptable for publication in the current form.